# Experimental Evaluation of an SDR-Based UAV Localization System

**DOI:** 10.3390/s24092789

**Published:** 2024-04-27

**Authors:** Cristian Codău, Rareș-Călin Buta, Andra Păstrăv, Paul Dolea, Tudor Palade, Emanuel Puschita

**Affiliations:** Communications Department, Technical University of Cluj-Napoca, 28 Memorandumului Street, 400114 Cluj-Napoca, Romania; cristian.codau@com.utcluj.ro (C.C.); rares.buta@com.utcluj.ro (R.-C.B.); andra.pastrav@com.utcluj.ro (A.P.); paul.dolea@com.utcluj.ro (P.D.); tudor.palade@com.utcluj.ro (T.P.)

**Keywords:** SDR, localization, angle of arrival, UAV communications, MUSIC algorithm, signal processing, direction-finding, uniform circular array

## Abstract

UAV communications have seen a rapid rise in the last few years. The drone class of UAV has particularly become more widespread around the world, and illicit behavior using drones has become a problem. Therefore, localization, tracking, and even taking control of drones have also gained interest. Knowing the frequency of a target signal, its position can be determined (as the angle of arrival with respect to a fixed receiver point) using radio frequency-based localization techniques. One such technique is represented by the subspace-based algorithms that offer highly accurate results. This paper presents the implementation of the MUSIC algorithm on an SDR-based system using a uniform circular antenna array and its experimental evaluation in relevant outdoor environments for drone localization. The results show the capability of the system to indicate the AoA of the target signal. The results are compared with the actual direction computed from the log files of the drone application and validated with a professional direction-finding solution (i.e., Narda SignalShark equipped with the automatic direction-finding antenna).

## 1. Introduction

Knowing the position of a target has always been a subject of interest with applications in the military, space, industrial, and civil domains. Target localization is used in enemy aircraft and spaceship localization and tracking, space surveillance and tracking, commercial flight tracking, or interference localization. Localization is also used in mobile communications, sonar systems, rescue operations in emergency scenarios in remote spaces, and so on.

The main methods used for localization are based on optical measurements/video, audio, sonar, and radio frequency (RF) communications. The most common RF localization systems use radar, interferometry, angle-of-arrival (AoA), ultra-wideband (UWB), or Wi-Fi positioning.

Unmanned aerial vehicle (UAV) communications have seen a rapid rise in the last few years. Commercial drones have become more widespread due to their ease of use, agility, and non-prohibitive prices. Applications vary from own-use photography and entertainment to crop monitoring for agriculture, mapping, delivery services, disaster relief, search and rescue, mining, or simply drone surveying. The exceptional maneuverability capabilities and the possibility to access remote places make drones a target for illicit behavior. Drug or contraband substance or gun smuggling, striking fire, reckless utilization, or simply unauthorized operations with drones need to be countered. Ongoing research tries to come up with solutions to address this problem. 

Therefore, given the importance of target localization and the necessity to counteract the unlawful operation of commercial drones, this paper presents a software-defined radio (SDR)-based system that can perform RF localization of drones. The proposed system uses a five-element uniform circular array (UCA) antenna and the multiple signal classification (MUSIC) subspace-based algorithm to identify the AoA of a commercial drone that operates in a known frequency band. While most literature works present the capabilities of this algorithm and its improvements using simulation, our work goes one step further and presents the experimental evaluation in relevant outdoor environments as well, showing the ability of the system to indicate the AoA of a target up to 2.5 km away. 

The rest of this paper is structured as follows: Section 2 presents the current state of the research in the field of drone localization. Section 3 describes the system model and the mathematical apparatus of the MUSIC algorithm for linear and circular array configurations. Section 4 discusses the algorithm simulation results and Section 5 addresses the algorithm implementation and the experimental testbed. Section 6 presents the measurement results compared against the real drone direction and the results of a professional direction-finding (DF) solution (i.e., Narda SignalShark equipped with the automatic direction-finding antenna, Narda ADFA, (Pfullingen, Germany)) for validation. Finally, Section 7 offers future perspectives and concludes the paper.

## 2. State of the Art in Drone Localization

In [1], the authors present a novel localization and activity classification method for UAVs based on mmWave frequency-modulated continuous-wave (FMCW) radar. These radars can detect targets located from 0.2 m up to 300–400 m away. Some advantages are the small antenna size, the durability, and the possibility of operation in adverse weather conditions. The radar system presented in the paper consists of a Texas Instruments mmWave FMCW radar with three transmitters and four receivers in a fixed position. Due to the small cross-section of the target (a drone of size 322 × 242 × 84 mm), it can be detected up to approximately 10 m range. The root mean square error (RMSE) in the estimation of the height of the drone and the distance from the radar is approximately 50 cm and 20 cm, respectively. The authors conclude that their method is useful in small-scale aerial vehicle traffic management ground stations. 

Research in [2] presents an overview of the techniques for modulation classification and signal strength-based localization of amateur drones (ADrs) using surveillance drones (SDrs) that have passive RF sensing ability. A key aspect of this solution is to fly the SDrs at higher altitudes than the ADrs. The better propagation (due to higher signal-to-noise ratio, line of sight, LOS) at high altitudes yields high accuracy in the detection and localization of ADrs. The approximate location of an identified target drone is obtained by multilateration based on the received signal strength (RSS) at the SDrs. The simulation assumes three SDrs positioned as the vertices of an equilateral triangle. The authors show that the localization error is minimal when the SDr altitude is 800 m and the coverage radius is 1000 m. The distance between the SDrs also affects the localization accuracy. The authors conclude that they have obtained a 10-fold increase in the coverage radius and a 25 dB reduction of the minimum detectable power by flying the SDrs at the optimal altitude. Four times better localization accuracy is also obtained by the optimum altitude of the SDrs. 

Paper [3] also proposes the deployment of monitoring drones (MDrs) for surveillance, localization, hunting, and jamming of ADrs. A flying ad hoc network (FANET) of MDrs can locate the position of the ADrs more accurately than a single MDr.

In [4], the authors present a sparse denoising autoencoder (SDAE)-based deep neural network (DNN) for direction finding of small UAVs. The system comprises a four-element directional antenna array and a single-channel receiver. The received signal power is measured at each antenna using an RF switching mechanism, and the SDAE-based DNN classifies/determines the direction of the drone signal. The proposed method was validated experimentally in an open area. The drone downlink signals are 10 MHz bandwidth OFDM signals that occupy the bandwidth from 2.401 GHz to 2.481 GHz. The space around the antenna system is divided into eight sectors, and the direction of the drone is indicated by the corresponding octant. 

In [5], the authors present the development of an anti-drone system that combines audio, video, and RF surveillance to realize drone detection, localization, and RF jamming. The experimental results show that the system can detect and localize an intruding drone in a campus environment. The authors also discuss the challenges of such systems. 

The paper in [6] presents a comprehensive overview of counter-drone systems, used to detect, localize, track, and neutralize UAVs. The strengths and limitations of sensors such as acoustic, RF, radar, electro-optical /infrared (EO/IR), and Light Detection and Ranging (LiDAR) are thoroughly described. Ground and sky platforms are compared, being suitable choices depending on the specific requirements of the applications.

The article in [7] provides another overview of the techniques used for detecting, localizing, and tracking unauthorized UAS and jammers, focusing on mmWave radar-, UWB radar-, and NLOS radar-based approaches. The authors also propose to use multiple UAS to localize and track another UAS. 

The approach in [8] is to detect and localize ultralight aircraft and drones using a WiFi-based passive radar for short-range surveillance applications. The advantages of passive radar compared with active radar are highlighted. The obtained results show that this approach can correctly detect and localize in 3D small flying objects along their trajectory, for short distances (approx. 100 m).

In [9], the authors demonstrate that W-band can be used to detect and localize small drones in 3D, using a FMCW radar. 

In [10], the authors present a novel algorithm for indoor localization of UAVs based on RSSI, with Wi-Fi access points and a priori known locations. 

The authors of [11] present the results of the detection and localization of drones using mmWave automotive radar sensor at 76–81 GHz. Given the limitations of the radar hardware in terms of transmitted power and the limitations in azimuth and elevation coverages by the 3-dB beamwidth, the range of detection and localization is up to 40 m. 

The paper in [12] presents a comprehensive overview of wireless target localization, such as time difference of arrival (TDOA), time of arrival (TOA), RSS, and AoA, showing their advantages and disadvantages. This review shows that the wireless target localization solutions available in the literature range between a few and hundreds of meters, depending on the approach and costs. 

In [13], the authors propose and experimentally validate an RF-based location-finding system for drones and their controllers. The system comprises SDRs and rotating antennas that combine RF-based drone detection with AoA triangulation. The drone used in the test flies 20 m above the ground and 30–150 m away from the system. The drone and its controller operate at a predefined 2.4 GHz Wi-Fi frequency channel. When monitoring the drone, the average error is 12.2° for direction finding and 12.71 m for localization, while for the controller, the average error is 9.9° for direction finding and 11.36 m for localization.

In [14], the authors present a 3D drone location estimation method using a 4-by-4 rectangular array and MUSIC algorithm. After the estimation of the AoA, an extended Kalman filter (EKF) is applied to improve the accuracy and to track the drone. The experiment takes place in an area of 10.4 × 20 m^2^, with the receiving system placed at the origin. The drone follows a 20 m flight trajectory in this area and continuously sends signals with an omnidirectional antenna. The obtained results are compared with the GPS coordinates. The mean error is 1.8 m, 2.6 m, and 0.9 m in the x, y, and z directions, respectively. 

The paper in [15] presents an evaluation and assessment of four AoA algorithms in UAV communication networks using multiple-input multiple-output (MIMO) base stations. The study shows that accurate AoA estimation can be achieved with as few as 320 samples. The number of antennas and their configurations impact the AoA estimations. The experiment is as follows: The base station is placed 34 m above the ground, in the stands of a stadium. A drone is placed at five positions, at angles of −9.64°, 10.20°, 21.54°, 32.49°, and 44.26°, at a height of 20 m, covering the pitch. The drone is kept in place for 17.92 s at each position, transmitting pilot signals with a 3.6 GHz center frequency, 5 MHz bandwidth, and 5 MHz sample rate. The base station collects 2.240.000 samples for every measurement point. Data are collected using eight planar array elements with a spacing of 3.94 cm. A comprehensive comparative analysis of five AoA algorithms is performed. Root-MUSIC offers the best results, with an inaccuracy from 3.54° to −5.88°, with a median of −1.63°. The paper shows that an increase in the number of antennas improves the accuracy of the results. It is also found that the effect of multipath propagation is more pronounced if the number of antennas is reduced. Moreover, it shows that uniform rectangular array (URA) configurations are better than uniform linear arrays (ULAs). By using multiple rows for their spatial components, the range of estimations can be reduced, and better overall accuracy can be obtained. Finally, the study shows that the 2D Bartlett spatial spectrum estimator offers better accuracy for both azimuth and elevation results than the 1D method, by aggregating all the antennas to locate the peaks in the spectrum. 

In [16], a frequency-hopping spread spectrum (FHSS) signal from a drone controller is detected and localized. After the controller is detected, two variants of the MUSIC algorithm are implemented to detect the AoA. The system comprises a four-channel receiver with quasi-Yagi antennas placed in a ULA configuration. The experiment is performed for five locations of the drone controller: 4.43°, 6.41°, 8.55°, 10.33°, and 2.1° for 115.24 m, 164.74 m, 214.74 m, 264.62 m, and 512 m, respectively. The results indicate an average error of 1.39°.

In [17], the authors present a 24 GHz ISM band FMCW radar system for drone localization designed to operate up to 150 m.

Another state-of-the-art solution is the use of MIMO radars employing electromagnetic vector sensors (EVS) [18,19,20] for 2D and 3D UAV localization. The work in [20] proposes the implementation of a monostatic EVS MIMO radar, while [19] models a bistatic configuration. In each case, an EVS is a collocated, complete polarized sensor system of three electric dipoles and three magnetic loops. The numerical simulation results show that a bistatic configuration is more reliable. Moreover, these approaches are insensitive to the geometry of the TX/RX array. 

The current state of the art reveals different approaches, architectures, and platforms used for drone localization using RF methods. Advantages and disadvantages corresponding to various scenarios or applications can be drawn: The radar implementations can detect targets up to a few hundred meters away, can be used mainly for short-range surveillance applications, have the advantage of small antenna sizes and durability, and can operate in adverse weather conditions. However, a large radar cross-section of the target is desired, which can be a problem for localizing small drones. Also, interferences from other small objects may affect the results, and line-of-sight operation is highly desired. RSS-based localization methods have an accuracy of tens of meters and can be considered cost-effective as most receivers can measure the RSS. The limitations are due to the noise and multipath and due to the requirement to know the target transmitted power. On the other hand, AoA-based localization can provide more accurate results but at the cost of more complex hardware, the necessity of synchronizing the receivers, and complex signal processing. Also, the signal frequency of the target must be known a priori. Moreover, in long-range scenarios, the SNR is lower than in close-range scenarios, which impacts the accuracy of the results. 

Extensive work is conducted to develop more complex mathematical models to better simulate the environment/behavior of the system and to propose new methods to improve AoA performance, such as signal sparse recovery (SSR), L1—singular value decomposition (SVD), nuclear norm optimization and sparse Bayesian learning (SBL), and optimal weighted subspace fitting (WSF) [21]. In [22], the authors model a complex practical environment, including nonuniform noise and off-grid errors, and propose an assistant vehicle localization method based on SBL-based robust DoA estimation and three collaborative base stations with MIMO arrays. Extensive simulations show that the proposed method provides superior localization results. Another complex model is presented in [21]. Here, a multi-UAV cooperative localization system for marine targets is proposed. As each UAV is equipped with a monostatic MIMO radar, the model considers the unknown mutual coupling effect and provides a robust weighted block sparse reconstruction DOA estimation method based on optimal WSF. In [23], the authors consider the mutual coupling and the reduced computational power of IoT sensors and propose a framework composed of distributed mobile-edge computing and IoT to realize auxiliary vehicle position analysis and develop a suitable block SBL algorithm for DF. 

While simulations provide valuable insights and initial assessments, it is crucial to acknowledge the inherent gap between simulation and reality when assessing and optimizing DF solutions. 

To this extent, we propose a practical solution based on the MUSIC AoA algorithm implemented on an SDR-based platform, which has several advantages: durability, small antenna size, possibility of operation in adverse weather conditions, high detection range, and high accuracy. However, similar to other AoA approaches, the performance is susceptible to non-LOS conditions and interferences, and we assume that the target frequency is known. The novelty of our work consists in the particular practical implementation and the validation of the proposed solution. As such, the main contributions of this study are as follows:Implementation of the MUSIC algorithm on an SDR-based system using a five-element UCA: MUSIC is a super-resolution algorithm and can achieve localization accuracy of up to a sub-degree. The use of a UCA with an odd number of elements ensures a non-ambiguous target localization in a full 360° range.Evaluation of the proposed solution capability to perform full azimuth (360-degree) RF signal localization: This is achieved in a close-range setup, in a controlled environment, using a dummy target signal.Performance evaluation in a relevant outdoor environment, in a long-range setup, for ranges up to 2.5 km using a drone as a target.Validation of the results by comparison with a professional DF solution (i.e., Narda ADFA) and by cross-referencing with the actual target AoA determined from the drone GPS coordinates.

## 3. The Mathematical Model of the MUSIC Algorithm

The MUSIC algorithm was proposed by R. Schmidt in 1979 [24]. MUSIC is a subspace-based algorithm that uses eigenvalue decomposition of the covariance matrix of the received signal at an antenna array to find the AoA of the signal [24,25,26].

Assume that there are D uncorrelated narrowband source signals, with the same central frequency, fc. These signals are impinging on an array of M elements that are linearly spaced with equal inter-element spacing d, equal to half wavelength. 

A data snapshot is the data received at all elements of the array at a single time t. Then, the received signal is given by *x*(*t*), as follows in (1):*x*(*t*) = *As*(*t*) + *n*(*t*),(1)
where *x*(*t*) is an M × 1 vector of received data consisting of signals and noise, *s*(*t*) is a D × 1 vector of source signal values from D sources, and *n*(*t*) is an M × 1 vector of noise at the array elements. Relation (2) shows the M × D matrix containing the steering vectors or arrival vectors for a ULA array configuration where the AoA is measured only for azimuth:*A* = [*a(θ*_1_*)/a(θ*_2_)*/…/a(θ_D_)*],(2)

A steering vector consists of the relative phase shift at the array elements of the plane wave from one source. Every column of matrix A is the steering vector from one of the sources and depends on the direction of arrival, *a*(*θ_D_*). The steering vector depends on the array configuration. For instance, considering a ULA array and taking the first element in the array as the reference element, the steering vector for direction (*θ_k_*) is given in (3) where symbol *^T^* stands for the transpose operation:*a*(*θ*_*k*_) = [1 *e*^(−*j*2*πd*/*λ*)*sin*(*θ*_*k*_^^)^ … *e*^(−*j*(*M*−1)2*πd*/*λ*)*sin*(*θ*_*k*_^^)^] ^*T*^(3)

In the case of a UCA array, the AoA is measured for both azimuth and elevation. The M × D matrix containing the steering vectors or arrival vectors [27] is shown in (4): *A* = [*a*(*φ*_1,_*θ*_1_)/*a*(*φ*_2,_*θ*_2_)/…/*a*(*φ*_*D*,_*θ*_*D*_)],(4)
where *φ_i_* ∈ [0, π/2] is the elevation angle and *θ_i_* ∈ [0, 2π] is the azimuth angle. The steering vector for direction *φ_j_*_,_*θ_k_* is given in (5):*a*(*φ*_*j*,_*θ*_*k*_) = [*e*^(*j*2*πr*/*λ*)*sinφ*_*j*_^^*cos*(*θ*_*k*_^^−*γ*_1_^^)^ *e*^(*j*2*πr*/*λ*)*sinφ*_*j*_^^*cos*(*θ*_*k*_^^−*γ*_2_^^)^…*e*^(*j*2*πr*/*λ*)*sinφ*_*j*_^^*cos*(*θ*_*k*_^^−*γ*_*M*−1_^^)^]^*T*^(5)
Here, *r* is the radius of the circle described by the UCA array elements and *γ_M_ = 2πm/M*.

As the algorithm takes uncorrelated noise into account, the covariance matrix is diagonal. It is found that the signal and noise subspaces are orthogonal to each other. 

When the signals are uncorrelated with the noise, the covariance matrix of the received signal has two components: the signal covariance matrix and the noise covariance matrix. Then, as shown in (6) where the symbol *^H^* represents the Hermitian operator and E is the expectation operator, we obtain:*R*_*x*_ = *E*{*xx^H^*} = *AR_s_A^H^* + *σ*_*n*_^2^*I*,(6)
where *R_s_* is the source covariance matrix, and *R_s_* = *E*{*ss^H^*}. *R_s_* is a positive-definite Hermitian matrix and has full rank D, equal to the number of sources (for uncorrelated sources or partially correlated sources). 

The signal covariance matrix, *AR_s_A^H^* has M × M dimension, with rank D < M. 

The noise power is considered to be equal at all sensors and uncorrelated between sensors. Therefore, the noise covariance matrix is an M × M diagonal matrix with equal values along the diagonal. Since the signal covariance matrix *AR_s_A^H^* has rank D, it means that it has D positive real eigenvalues. The eigenvectors corresponding to these D eigenvalues span the signal subspace, given in (7): *U_s_* = [*v*_1_*,…,v_d_*].(7)

The M-D eigenvalues correspond to the noise subspace, and their eigenvectors span the null subspace, given in (8):*U_n_* = [*u*_*d*+1_*,…,u_m_*].(8)

The MUSIC algorithm exploits the orthogonality relationship between the signal and noise subspace. Then, as shown in (9), it results in: *A^H^u_i_ =* 0*.*
(9)

Therefore, the arrival vectors are orthogonal to the noise subspace. 

The MUSIC algorithm searches for all arrival vectors that are orthogonal to the noise subspace by computing the so-called MUSIC pseudospectrum using Relation (10): *P_MUSIC_ (φ) =* 1*/a^H^(φ)U_n_U_n_^H^a(φ)*
(10)

The above equation results in high peaks (theoretically infinite) that correspond to the desired directions of arrival. The pseudospectrum can have more peaks than there are sources, so the number of sources must be specified as a parameter, i.e., P. The algorithm will return the P largest peaks. The number of sources must be smaller than the number of array elements. The estimation efficiency of the MUSIC algorithm depends on the spacing between the array elements and the number of elements. The maximum efficiency is obtained for an inter-element spacing of half-wavelength of the operating frequency. Increasing the number of elements in the array also gives better results (sharper peaks in the MUSIC pseudospectrum).

## 4. MUSIC Algorithm Simulation

Matlab simulations prove the capabilities of the MUSIC algorithm to identify the AoA of incident signals impinging on an array of antennas. The most common array is the ULA array. As mentioned in the previous section, the best results are obtained for an inter-element spacing of half lambda of the operating frequency. Increasing the number of antennas not only yields sharper peaks in the pseudospectrum but also allows for more signals to be located as the number of sources must be smaller than the number of array elements. Two signals impinging on a ULA array of five elements spaced at half lambda are simulated: The first signal is from −57°, and the second signal is from 37°. The signal-to-noise ratio (SNR) is 5 dB. 

Figure 1 shows the MUSIC pseudospectrum of the received signals at the array elements. 

The results in Figure 1 clearly show the ability of the MUSIC algorithm to correctly identify the AoA of the two signals. However, the ULA array configuration has some drawbacks in detecting the AoA of incident signals. First, in a ULA setup, the orientation determines the reference direction; therefore, it cannot distinguish between signals arriving from the front and those arriving from the back of the array (front-to-back ambiguity). Second, the ULA configuration may exhibit inaccuracies in the end-fire region, when the target is in line with the elements of the array, leading to insufficient angular diversity (end-fire ambiguity). As a result, the algorithm may produce erroneous results or fail to provide reliable direction estimates. Third, a ULA array can only evaluate the AoA in azimuth.

To overcome the limitations of ULA, a UCA configuration can be used. The circular geometry offers multiple advantages. First, the UCA can cover the entire azimuthal plane (full 360° range), ensuring that targets can be localized regardless of their angle of arrival relative to the array. Second, the UCA does not suffer from front-to-back and end-fire region ambiguity, providing unambiguous direction estimation. Third, UCA can be used to estimate both azimuth and elevation at the same time by careful consideration of the array design and signal processing.

Two signals impinging on a five-element UCA with inter-element spacing of half lambda are simulated. The first signal comes from [−57° azimuth; 50° elevation] and the second signal [37° azimuth; 20° elevation]. The SNR is 5 dB.

Figure 2 illustrates the results of the MUSIC algorithm.

Figure 2 shows the ability of the UCA configuration to identify the AoA, in terms of both azimuth and elevation, of two incident signals. 

Following the simulation results, in the next section, the MUSIC algorithm implementation and the experimental setup are presented.

## 5. System Architecture and Algorithm Implementation

The system comprises the following elements: (1)Three National Instruments Universal Software Radio Peripheral 2954R (NI USRP) SDRs (Austin, TX, USA);(2)One OctoClock CDA-2990 clock distribution device (National Instruments, Austin, TX, USA);(3)One CPS-8910 switch (National Instruments, Austin, TX, USA);(4)One NI PXIe-8880 host computer (National Instruments, Austin, TX, USA);(5)One Hameg HM 8135 signal generator (now Rohde & Schwarz, Munich, Germany);(6)Five L-com omnidirectional antennas (L-Com, North Andover, MA, USA).

The NI USRP 2954R SDR [28] is a reconfigurable SDR device equipped with a Xilinx Kintex-7 FPGA. The frequency range is from 10 MHz to 6 GHz with a maximum instantaneous real-time bandwidth of 160 MHz and a maximum I/Q sample rate of 200 MS/s. The resolution of the analog-to-digital (ADC) converter is 14 bits. 

The three SDRs receive the target signals, which are then processed by the host computer. The SDRs are controlled by the host computer via the CPS-8910 switch. The connection between the SDRs and the switch, and the switch and the host computer, are made with PCIe x4 cables and a PCIe ×8 cable, respectively. 

The NI PXIe-8880 host computer has an 8-core Intel Xeon CPU E5-2618L v3@2.3 GHz processor and 24 GB RAM. In addition to controlling the SDRs, the computer also runs the software for storing and processing the received data, which are streamed from the SDRs. First, LabVIEW Communications 2.0 (product of NI) is used to store the received samples on the computer. Then, these samples are processed (offline) in Matlab R2020a (Natick, MA, USA) where the MUSIC algorithm is implemented. 

The OctoClock CDA-2990 [29] and the Hameg HM 8135 [30] are used to synchronize the receiving channels of the proposed solution for coherent operation as DF applications require time, frequency, and phase synchronization. The OctoClock is a high-accuracy time and frequency reference distribution device and provides 1PPs and 10 MHz reference signals for time and frequency synchronization. For most receivers, the digital down converter (DDC) chain uses a coordinate rotation digital computer (CORDIC). The CORDIC has a random start-up position on power-up that creates a random phase each time the channels of the receiver are initialized but remains constant through operation [4,31]. This means that a calibration procedure is necessary to find and compensate for the random phase shift that appears on every RF channel. In our system, the Hameg signal generator is used to transmit the calibration signal for each of the five RF channels to achieve phase alignment. 

The L-com omnidirectional antennas [32] are connected to the RF input ports of the three USRPs (one USRP has 2 TX/RX RF ports). The antennas are placed in a UCA array configuration. The distance between consecutive elements of the array is half of the wavelength corresponding to 2.46 GHz. 

Figure 3 illustrates the conceptual architecture of the system. 

Figure 4 presents the implementation of the conceptual architecture using NI hardware components.

The parameters for all five receiving channels are set in the LabVIEW environment as follows: central frequency, sampling frequency, number of samples, and gain. 

The implemented AoA method has 4 steps: (1) calibration samples acquisition, (2) DF samples acquisition, (3) calibration, and (4) AoA computation. 

(1) Calibration samples acquisition—The signal generator is set to prepare the calibration tone (a sine tone on the central frequency of the receivers). The output port of the signal generator is connected to an antenna placed in the center of the circle described by the five antennas of the UCA array. Thus, the signal arrives at the antennas at the same time. The phase differences obtained between the array elements are due to the random phase shift that occurs at retune commands and is constant during operation. Then, the receivers and the signal generator are set to run. The SDRs stream the received calibration samples to the host computer. A LabVIEW application stores these samples in binary files. After that, the signal generator is removed. 

(2) DF samples acquisition—The SDRs stream the received DF samples to the computer. These samples are also stored in binary files for offline processing.

(3) Calibration—In a Matlab application, the calibration procedure is performed based on the calibration samples and DF sample files. The phase differences obtained from the calibration files are then used in the DF samples to achieve the necessary coherent operation of the five channels. 

(4) Finally, the MUSIC algorithm can be applied to obtain the AoA of the desired signals. This is achieved using the Matlab phased array system toolbox.

## 6. Measurement Results

The performance of the system is evaluated in two scenarios:Scenario 1—Close-range experiments: A signal generator transmits a continuous sine wave as the target signal. The target is placed in 36 close-range (three-meter) positions to test full azimuth signal localization.Scenario 2—Long-range experiments: A drone is used as a target. The drone is flown up to 2.5 km from the receiving system.

### 6.1. Scenario 1—Close-Range Experiments

In this scenario, the target is a continuous sine wave from a signal generator. The target is placed in 36 positions that describe a circle of a 3 m radius. The positions are at the angles from 0 to 360° with respect to the reference of the receiving system, as shown in Figure 5. The height of the target is approximately the same as the height of the UCA array, 1 m above the ground. 

Figure 6 shows the experimental setup.

The frequency of the signal is 2.46 GHz. The I/Q rate of the receivers is 1MSample/s. A snapshot contains 5000 complex I/Q samples, and a file contains 250 snapshots. In this scenario, for every position of the target, 250 AoA values are computed using the MUSIC algorithm (only azimuth). Histogram representation is used to display the results. Figure 7 illustrates the MUSIC pseudospectrum of one snapshot of the target signal at 30°. 

Unlike the simulated pseudospectrum, which exhibited high, narrow peaks only for the target sources, the MUSIC pseudospectrum obtained during the measurement campaigns is slightly different. Although there is only one source that is accounted for (our target signal), the pseudospectrum shows several peaks of various amplitudes. A five-element array can indicate up to four target AoAs (number of array elements—1). Here, the highest peak corresponds to the dummy target (AoA of 38°) while the other peaks indicate interfering sources or reflections. These differences occur because numerical simulations consider idealized conditions while real-world experiments are affected by the inherent complexities and uncertainties of the environment. Noise, variability of the propagation environment, and unknown interference sources can significantly impact the results. Figure 8 shows the histogram based on the 250 AoA measurements for the target at 45°. 

The histogram representation in Figure 8 shows the AoA mode (i.e., the value that appears most frequently) with its frequency. The AoA value 45° appears 105 times from the total 250 results. The low standard deviation of the 250 measured values can also be seen, showing the consistency of the AoA determinations. 

The results for Scenario 1 are presented in Table 1. For each position of the target signal, the following statistical data are shown: the AoA value that appears most frequently in the histogram and its count, the mean value, the standard deviation, and the error from the actual position of the target. These data are computed after eliminating the outliers. The percentage of outliers is also indicated in the table. 

As per theoretical expectations, the AoA results for the 36 positions show that the proposed system, using MUSIC and the five-element UCA configuration, can be used for 360° AoA localization of a target. Due to the UCA geometry and odd number of array elements, the results show no front-to-back and end-fire region ambiguities. It should also be noted that the resolution of the scan angles is one degree, meaning that the resolution of the AoA determination is also one degree. As such, the system can clearly distinguish between the 36 target positions. The overall average error (with respect to the position of the target) is 3.2713°. In particular, the highest error is 7.8354° (target position at 10°). The lowest error is 0.1673° (target position at 290°). A possible source of error is the inaccuracy of the physical placement of the target. 

The overall average percentage of outliers is 3.81%. The highest number of outliers for a given position of the target is 32, or 12.8% (target at 40°). For several positions, the results show no outliers. The low number of outliers can mean that there were few or no interfering signals in the received and processed samples. 

The overall average standard deviation is 1.0883°. This value indicates the consistency of the obtained results along all measurements. Figure 9, Figure 10, Figure 11 and Figure 12 show the histograms for the target positions from 0° to 90°, 90° to 180°, 180° to 270°, and 270° to 360°, respectively. μ represents the AoA mean value and σ is the standard deviation.

### 6.2. Scenario 2—Long-Range Experiments

In the second scenario, a DJI Mavic 3T-Basic Enterprise drone (DJI, Shenzhen, China) [33] is used as a target. This drone is a popular commercial drone and is a suitable choice for our experiments for several reasons. It has a maximum flight time of up to 45 min and a maximum transmission range of up to 15 km (FCC) in LOS, which allows us to perform a high number of measurements at various points across a large area in a single flight. The drone uses the 2.4 GHz and the 5.8 GHz ISM frequency bands with user-configurable DL frequency and bandwidth, which allows us to set the target frequency and match it with the phase calibration signal. Moreover, the drone provides log files with GPS flight data, which allows us to compute the actual AoA of the drone and use it as a reference when evaluating the system’s performance.

In Scenario 2, the downlink signal is an OFDM signal with a bandwidth of 20 MHz, centered on 2.46 GHz. The drone is flown from 125 m to 2.5 km away from the receiving system in LOS, at an altitude of 300 m, hovering in 14 distinct positions. AoA measurements (azimuth) are made for the 14 positions of the drone using two different solutions: (1) the proposed system and (2) a professional DF solution (i.e., Narda ADFA) [34]. These positions are marked with the dark red points on the map presented in Figure 13. 

The actual AoA angles are measured with respect to the ‘North’ orientation of the Narda ADFA, as illustrated in Figure 14. The yellow mark is the reference of the system.

Figure 15 shows the receiving system.

The receiver parameters are the same as in the first scenario: the I/Q rate is 1MSample/s, a snapshot contains 5000 complex I/Q samples, and a file contains 250 snapshots. For every position of the target 250 AoA values are computed using the MUSIC algorithm. 

The results for Scenario 2 are presented in Table 2. For each position of the target signal, the following statistical data are shown: the AoA value that appears most frequently in the histogram and its count, the mean value, the standard deviation, and the error from the actual position of the target. These data are computed after eliminating the outliers. The percentage of outliers is also indicated in Table 2.

Figure 16 presents a comparison between the obtained results. The X-axis shows the 14 positions while the Y-axis displays the corresponding AoA value (measured by the Narda ADFA, the proposed system, and calculated based on the GPS positions).

These results show that the proposed system can indicate the AoA of a target that is up to 2.5 km away. Compared with the short-range scenario, the long-range scenario had more erroneous results, proving the theoretical construct that a low SNR decreases the accuracy of the AoA estimation. In the short-range scenario, the SNR is almost constant, being controlled as the target signal is a continuous wave of constant amplitude and the distance is constant (3 m). However, in the long-range scenario, the SNR varied uncontrollably because of the changing distance (125–2500 m) and the drone’s adaptive power feature. 

The average error with respect to the AoA computed based on the drone log files is 18.65°. The minimum error is 12.16°, while the maximum error is 25.92°. For all 14 positions, the MUSIC implementation gives higher values than the GPS-based data. This can be caused by slightly different origins of the two systems and slightly different North references. The average error with respect to the Narda ADFA professional solution is 4.27°. The minimum error is 0.4835°, and the maximum error is −11.96°. The two receivers do not have the same origin (as can be seen in Figure 14), hence a deviation between the results of the measurements is understandable. The over-the-air calibration can also contribute to a constant error. Ideally, the calibration signal should be fed to the receiving channels via cables of the same length to estimate the phase shifts as reliably as possible. 

The overall average percentage of outliers is 1.75%. The highest number of outliers for a given position is 24 (drone at 1 km, 128.32°). There are several positions where there are no outliers. The low percentage of outliers can mean that there are no interferences in the received, collected, and processed samples. The average AoA standard deviation is 0.9878. This low deviation indicates the high precision of the results throughout the experiments. 

The measurement results from both scenarios show that the proposed system can evaluate the AoA of incoming signals impinging on the UCA array. In both cases, the results show a low percentage of outliers in the AoA values, which can be linked to little or no interferences. In the first scenario, since the target signal is a continuous tone and it is in close range to the receiver, it is unlikely that interferences override the target signal. In the second scenario, however, the target is the downlink signal from a drone. This signal is not continuous as it has a gap of a few milliseconds between packets, like WLAN signals. Thus, if the received snapshots correspond to the gaps between the packets, the resulting AoA value is not relevant. 

## 7. Conclusions

The rise of UAV communications leads to a corresponding rise in safety measures associated with it. Drone localization is, therefore, a hot topic and can be performed in a variety of methods depending on the available resources. An SDR-based UAV localization system is presented in this paper. The proposed system implements the super-resolution MUSIC algorithm using a five-element UCA array. The received samples are recorded and stored in files using LabVIEW. These files are then processed offline in Matlab. The performance of the system is evaluated in two outdoor scenarios. The first scenario tests the ability to perform 360° localization in close range using a continuous signal as the target. The second scenario tests the long-range ability to localize the AoA of a drone with a known operating frequency that is flown to 2.5 km from the receiving system. The results show the capability of the proposed solution to localize the AoA of RF targets up to 2.5 km. To validate the proposed solution, a professional direction-finding system (i.e., Narda ADFA) is also used in the second scenario, yielding similar results. The low standard deviations for both scenarios indicate the high precision of the obtained AoA results when there is no interference. 

In case of interference, several major sets of concentrated values appear on the histogram representation, one of them corresponding to the target. This means that even though there is interference, the target signal is still located, but, by analyzing these results, it cannot be said which is which. Moreover, including such results in a statistical analysis would yield high errors, high standard deviations, and various mean values, and would make it difficult to extract conclusive remarks. To overcome this in a real-time implementation, a heatmap of the obtained results is a suitable method for estimating and keeping track of the AoA of the target signal. 

Having a precise North reference of the array is important for obtaining accurate results. In our implementation, the 0° angle points toward the geographical North. As in many DF systems, this aligns with the concept of azimuth where angles are measured clockwise from the North direction in a horizontal plane. To achieve this, a compass can be used to properly align the UCA, but magnetic declination should be accounted for to obtain accurate results. A GPS receiver can also be used to determine the position and the orientation of the UCA. Moreover, once the UCA is properly aligned, the antennas should be anchored for proper operation in windy conditions. In future work, this aspect will be more rigorously addressed to remove the positioning errors and offsets caused by array misalignment. 

To further improve the solution, the authors aim to (1) implement real-time system operation by integrating all signal processing in a single application and (2) reduce the size of the system by implementing the AoA algorithm on an FPGA board with multiple coherent channels.

## Figures and Tables

**Figure 1 sensors-24-02789-f001:**
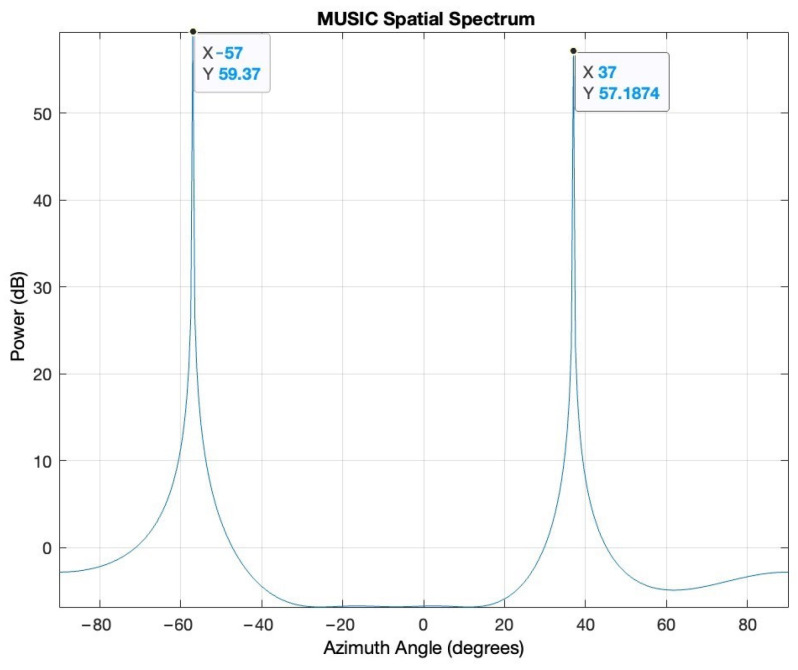
MUSIC simulation of the ULA array.

**Figure 2 sensors-24-02789-f002:**
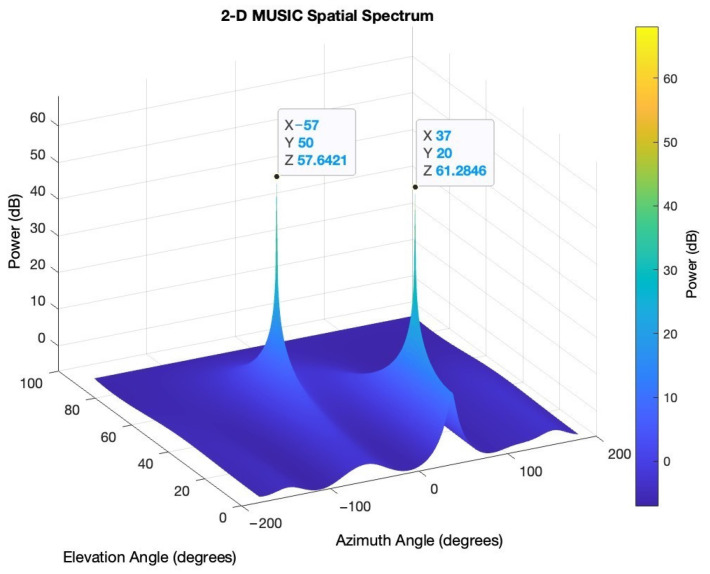
MUSIC simulation of the UCA array.

**Figure 3 sensors-24-02789-f003:**
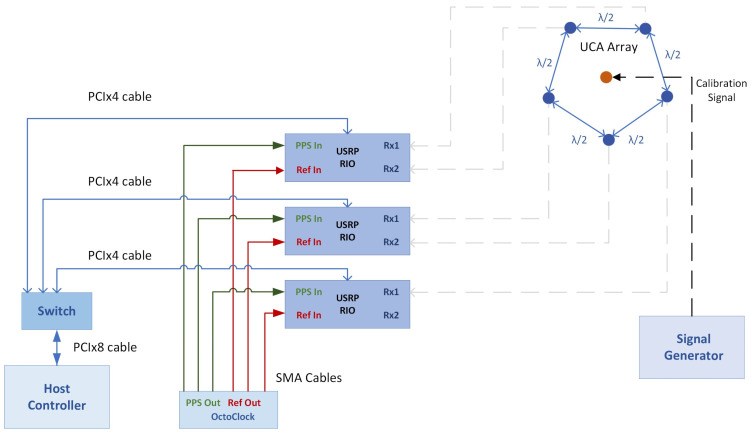
System conceptual architecture.

**Figure 4 sensors-24-02789-f004:**
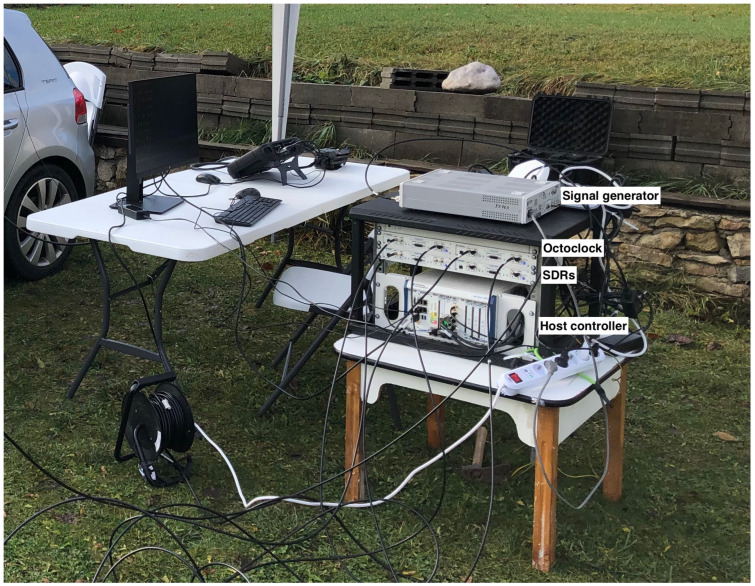
Implementation of the system architecture using hardware components.

**Figure 5 sensors-24-02789-f005:**
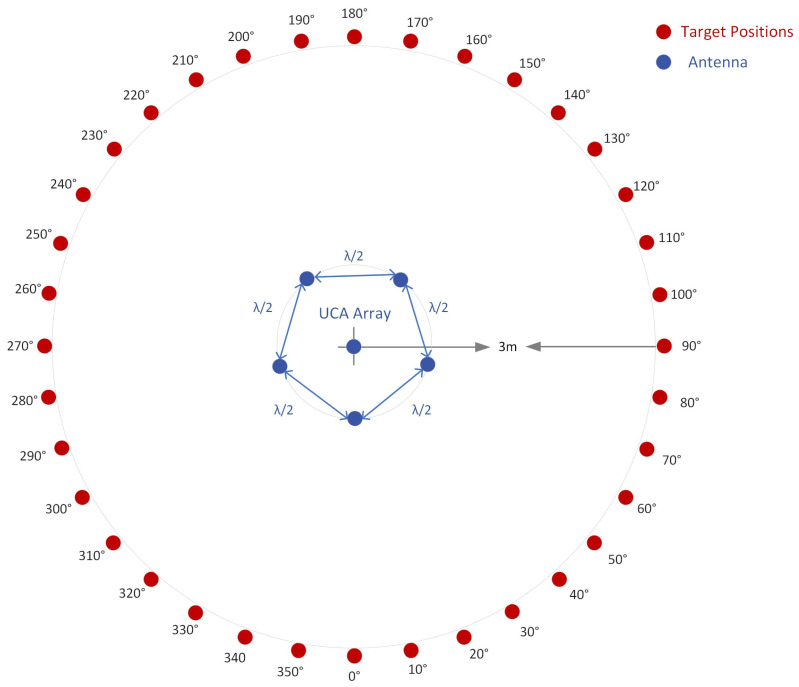
Target positions in Scenario 1.

**Figure 6 sensors-24-02789-f006:**
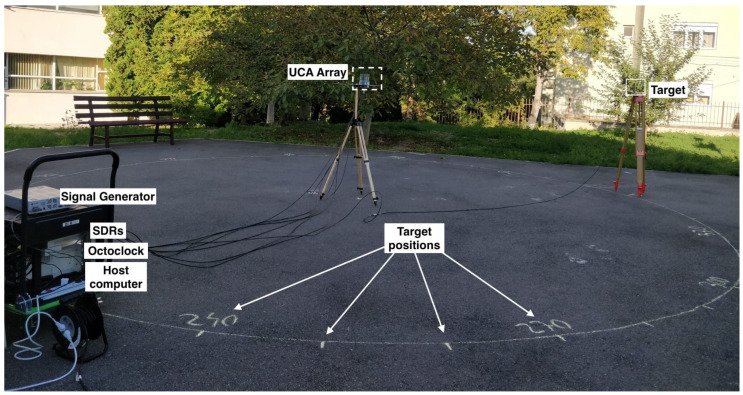
Experimental setup for Scenario 1.

**Figure 7 sensors-24-02789-f007:**
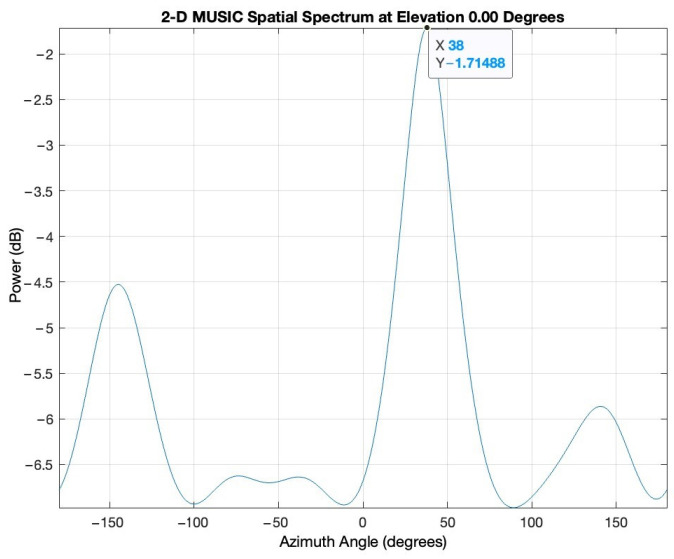
MUSIC pseudospectrum of one snapshot.

**Figure 8 sensors-24-02789-f008:**
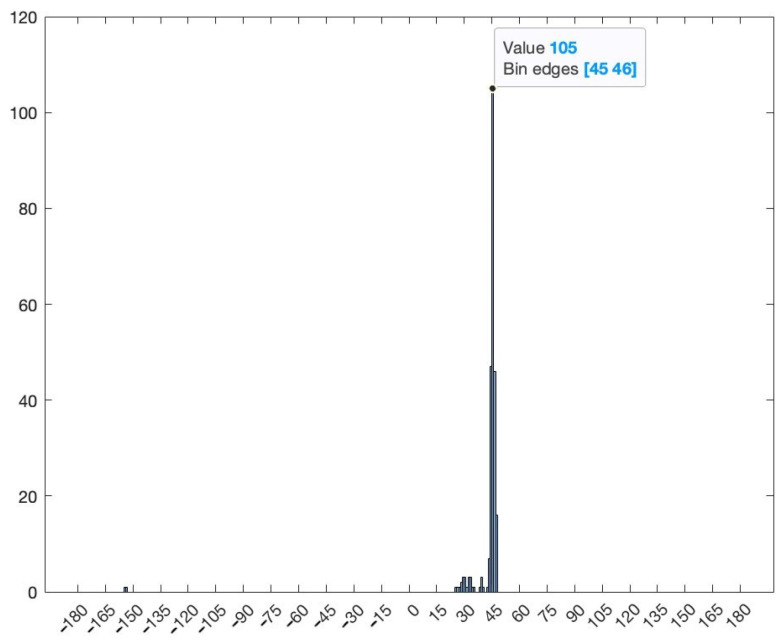
Histogram for one position of the target.

**Figure 9 sensors-24-02789-f009:**
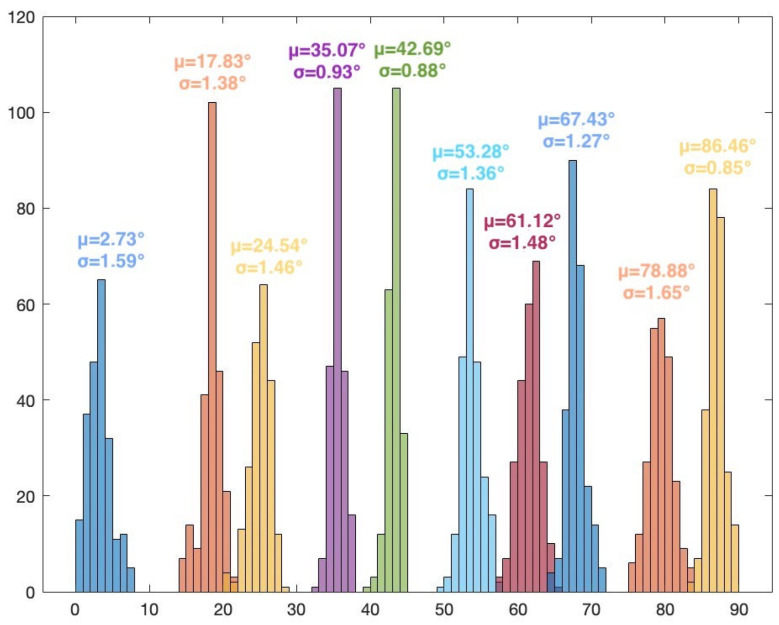
Histogram for target positions from 0° to 90°.

**Figure 10 sensors-24-02789-f010:**
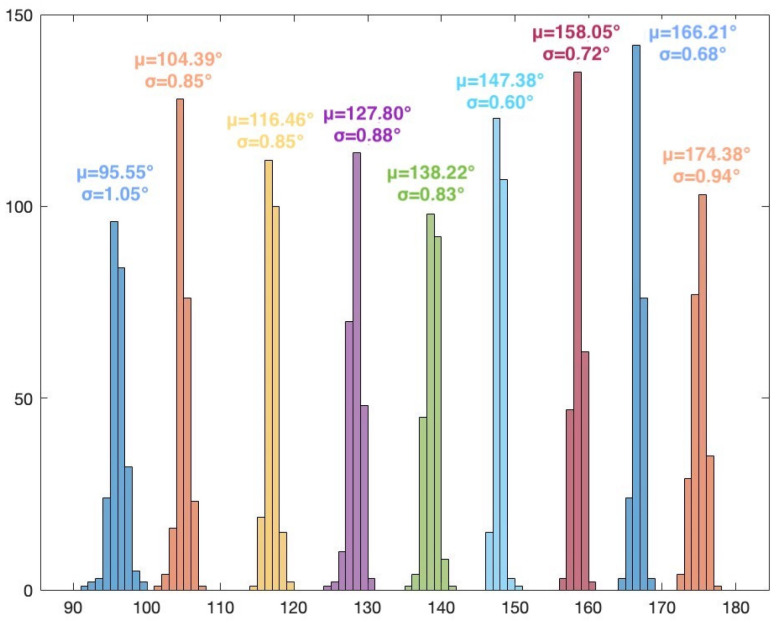
Histogram for target positions from 90° to 180°.

**Figure 11 sensors-24-02789-f011:**
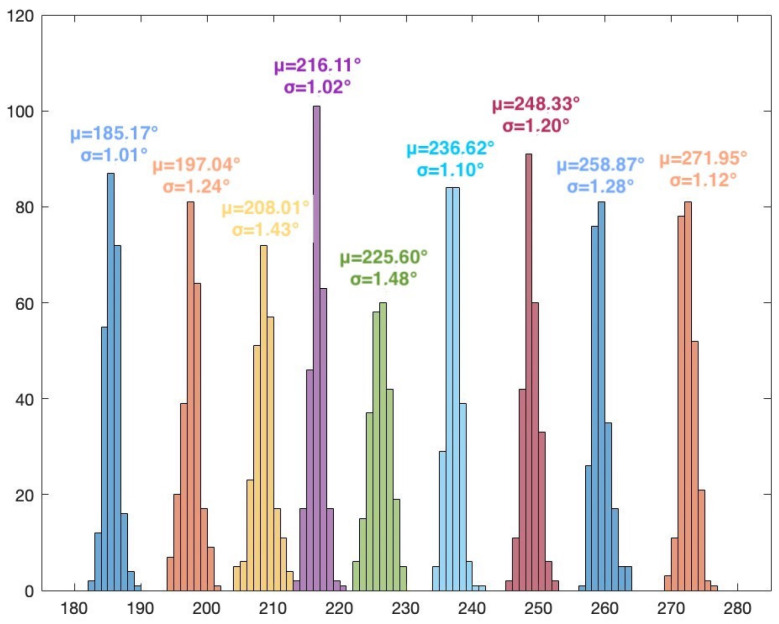
Histogram for target positions from 180° to 270°.

**Figure 12 sensors-24-02789-f012:**
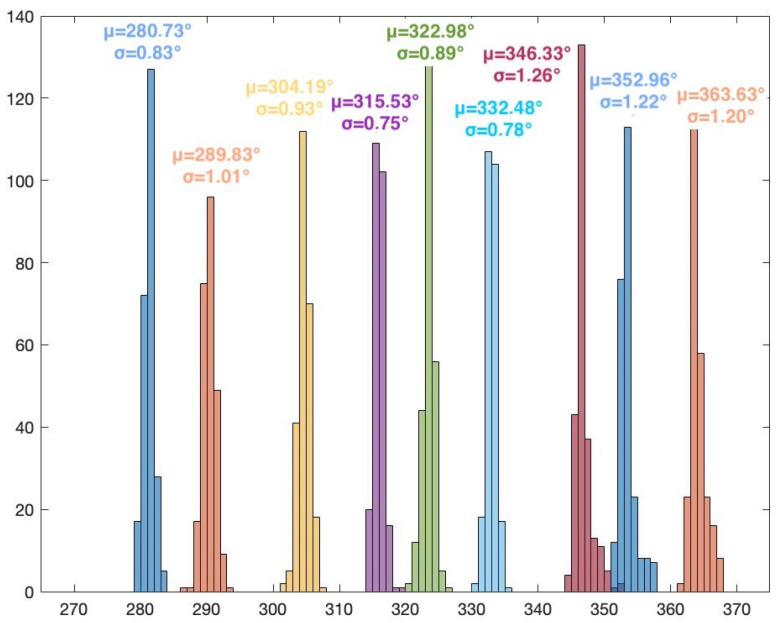
Histogram for target positions from 270° to 360°.

**Figure 13 sensors-24-02789-f013:**
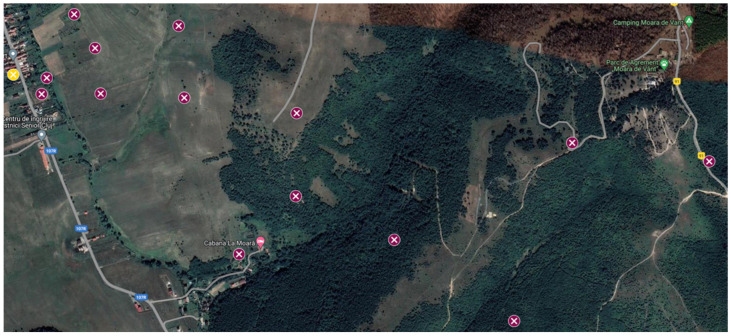
Map view of the measurement points.

**Figure 14 sensors-24-02789-f014:**
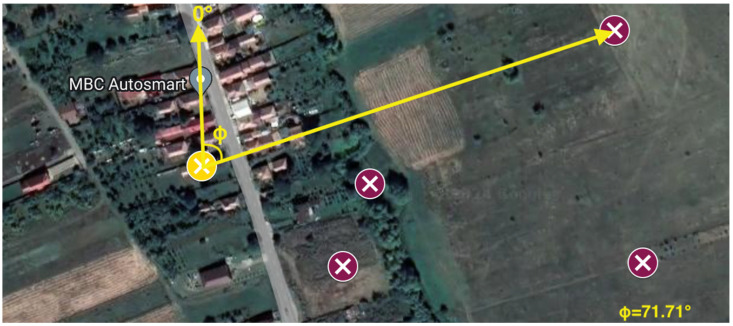
Actual AoA angles calculation based on the reference.

**Figure 15 sensors-24-02789-f015:**
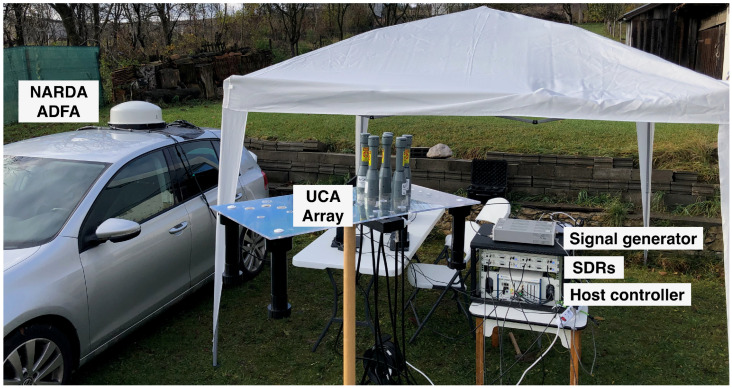
Receiving system for Scenario 2.

**Figure 16 sensors-24-02789-f016:**
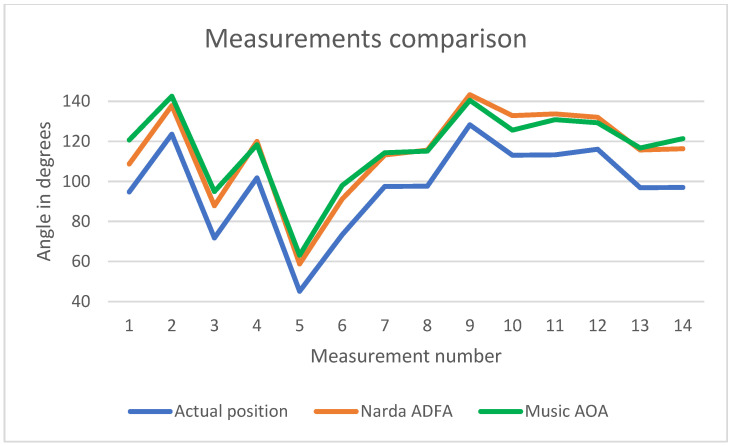
Measurement comparison.

**Table 1 sensors-24-02789-t001:** System performance in Scenario 1.

Target Position (°)	AoA Mode (°)	Mode Count	AoA Mean Value (°)	AoA Standard Deviation (°)	Mean Error (°)	Outliers (%)
0	3	65	2.7309	1.5909	2.7309	10.44
10	18	102	17.8354	1.3848	−7.8354	4.82
20	25	64	24.5413	1.4562	−4.5413	12.45
30	35	105	35.0679	0.9342	−5.0679	11.24
40	43	105	42.6912	0.8775	−2.6912	12.85
50	53	84	53.2762	1.3624	−3.2762	4.02
60	62	69	61.1224	1.4826	−1.1224	1.61
70	67	90	67.4256	1.2707	2.5744	2.81
80	79	57	78.8824	1.6493	1.1176	4.42
90	86	84	86.4653	1.1925	3.5347	1.61
100	95	96	95.5533	1.0469	4.4467	2.01
110	104	128	104.3936	0.8506	5.6064	0
120	116	112	116.4603	0.7597	3.5397	4.02
130	128	114	127.8050	0.8799	2.1950	3.21
140	138	98	138.2254	0.8280	1.7746	2.01
150	147	123	147.3836	0.6061	2.6164	6.83
160	158	135	158.0522	0.7193	1.9478	0
170	166	142	166.2129	0.6772	3.7871	0
180	175	103	174.5582	0.9406	5.4418	0
190	185	87	185.1739	1.0131	4.8261	7.63
200	197	81	197.045	1.2359	2.9550	10.84
210	208	72	208.0073	1.4303	1.9927	6.43
220	216	101	216.1116	1.0236	3.8884	6.43
230	226	60	225.605	1.4853	4.3950	4.42
240	237	84	236.6157	1.0956	3.3843	2.81
250	248	91	248.3293	1.2026	1.6707	1.20
260	259	81	258.8735	1.2757	1.1265	1.61
270	272	81	271.951	1.1188	−1.9510	1.61
280	281	127	280.7269	0.8265	−0.7269	0
290	290	96	289.8327	1.0084	0.1673	1.61
300	304	87	304.1885	0.9281	−4.1885	2.01
310	315	109	315.5336	0.7554	−5.5336	4.42
320	323	129	322.9799	0.8865	−2.9799	0
330	332	107	332.4798	0.7790	−2.4798	0.40
340	346	133	346.3293	1.2652	−6.3293	0
350	353	113	352.9628	1.2233	−2.9628	2.81
360	363	115	363.6337	1.2033	−3.6337	2.41

**Table 2 sensors-24-02789-t002:** System performance in Scenario 2.

GPS Data		Narda ADFA	AoA System Performance						
Distance (m)	Direction (°)	Narda DF (°)	AoA Mode (°)	Mode Count	AoA Mean Value (°)	AoA Standard Deviation (°)	Mean Error (°)		Outliers %
							vs. GPS Data	vs. Narda DF	
126.46	94.74	108.7	121	96	120.6571	0.8665	25.9171	−11.9571	1.61
128.43	123.62	137.9	142	133	142.5663	0.6695	18.9463	−4.6663	0.00
308.02	71.71	87.8	95	84	94.8988	1.3532	23.1888	−7.0988	0.80
320.11	101.8	120	118	72	118.2915	1.3867	16.4915	1.7085	0.80
314.13	45.15	58.8	63	64	63.1347	1.6920	17.9847	−4.3347	1.61
620.01	73.38	91.2	98	101	97.9878	0.8829	24.6078	−6.7878	1.20
618.57	97.45	113.2	115	82	114.3074	1.1758	16.8574	−1.1074	2.01
1021.92	97.59	115.6	116	110	115.1165	0.9538	17.5265	0.4835	0.00
1028.6	128.32	143.3	140	120	140.4800	0.6949	12.16	2.82	9.64
1107.01	113.09	132.8	126	105	125.5943	0.9704	12.5043	7.2057	2.01
1483.27	113.27	133.7	131	84	130.8115	0.9883	17.5415	2.8885	2.01
1985.69	116.05	132	129	176	129.2731	0.4812	13.2231	2.7269	0.40
2014.13	96.86	115.7	117	162	116.7028	0.5536	19.8428	−1.0028	0.00
2499.45	96.99	116.3	121	94	121.3292	1.1610	24.3392	−5.0292	2.41

## Data Availability

Raw data are not publicly available due to restrictions.

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
