# Peer review of "Experimental Evaluation of an SDR-Based UAV Localization System"

_sensors, 2024, doi:10.3390/s24092789_

Round 1

Reviewer 1 Report

Comments and Suggestions for Authors

In the Abstract you mention the professional solution (i.e., Narda Signal Shark equipped with the automatic direction-finding antenna). In the Introduction.
you mention "professional COTS Direction Finding (DF) solution". Please find the right words to explain if these are the same professional solution or different or part of the same spectrum. 

The Hameg Hm 8135 [25] is a signal generator that operates up to 3GHz. It is required for a “complete” synchronization of the five channels. Please bring some explanations about "complete". Is this supposed to be referring to pseudo-completeness or ? 

Measurement results. You present the scenarios very well but I would advise to change the name of scenarios from Scenario 1 to Scenario 1 - Close range and same for scenario 2.  

For scenario 2, you mention that you used DJI Mavic. For sure you didn't used this drone randomly. Please provide some observations, technical and application-oriented reasons why you picked this one.

In the conclusions you mention that "Having a precise North reference of the array is important ". Could you also indicate some possible ways that could be achieved or researched in the future work ? 

Author Response

Response to Reviewer 1 Comments

1. Summary

We sincerely thank the reviewer for taking the time to analyze our work and provide us with valuable advice on how to improve our research. Please find the detailed responses below and the corresponding revisions/corrections highlighted in blue in the revised manuscript.

2. Questions for General Evaluation

Reviewer’s Evaluation

Response and Revisions

Does the introduction provide sufficient background and include all relevant references?

Yes

Are all the cited references relevant to the research?

Yes

Is the research design appropriate?

Can be improved

Please see the point-by-point response section below

Are the methods adequately described?

Can be improved

Please see the point-by-point response section below

Are the results clearly presented?

Yes

Are the conclusions supported by the results?

Yes

3. Point-by-point response to Comments and Suggestions for Authors

Comments 1: In the Abstract you mention the professional solution (i.e., Narda Signal Shark equipped with the automatic direction-finding antenna). In the Introduction you mention "professional COTS Direction Finding (DF) solution". Please find the right words to explain if these are the same professional solution or different or part of the same spectrum. 

Response 1: Thank you for pointing this out. We have addressed this shortcoming by consistently referring to the professional solution as “professional direction finding solution (i.e., Narda Signal Shark equipped with the automatic direction-finding antenna)”, or in brief “Narda ADFA”. These changes occur in the revised manuscript in lines: 19-20, 59-61, 235-236, 476, 479-480, 495, 509 and 542-543.

Comments 2: The Hameg Hm 8135 [25] is a signal generator that operates up to 3GHz. It is required for a “complete” synchronization of the five channels. Please bring some explanations about "complete". Is this supposed to be referring to pseudo-completeness or ? 

Response 2: Thank you for bringing this to our attention. To operate properly, the DF solution requires to be perfectly synchronized in time, frequency, and phase – this was initially referred to as “complete” synchronization. Therefore, we use the OctoClock for time and frequency synchronization and the Hameg signal generator to generate a calibration signal to compensate for any initial random phase shift. This aspect was clarified in the revised manuscript in lines: 353-364, as follows: The OctoClock CDA-2990 [29] and the Hameg HM 8135 [30] are used to synchronize the receiving channels of the proposed solution for coherent operation, as DF applications require time, frequency and phase synchronization. The OctoClock is a high-accuracy time and frequency reference distribution device and provides 1PPs and 10MHz reference signals for time and frequency synchronization. For most receivers, the digital down converter (DDC) chain uses a coordinate rotation digital computer (CORDIC). The CORDIC has a random start-up position on power up, that creates a random phase each time the channels of the receiver are initialized, but remains constant through operation [31], [4]. This means that a calibration procedure is necessary to find out and compensate the random phase shift that appears on every RF channel. In our system, the Hameg signal generator is used to transmit the calibration signal for each of the five RF channels to achieve phase alignment.

Comments 3: Measurement results. You present the scenarios very well but I would advise to change the name of scenarios from Scenario 1 to Scenario 1 - Close range and same for scenario 2. 

Response 3: Thank you for your valuable advice. As per your suggestion, we have referred to scenario 1 as “close-range” and to scenario 2 as “long-range”. These changes occur in the revised manuscript in lines: 231, 233-234, 397, 400, 402, 461, 538, and 539.

Comments 4: For scenario 2, you mention that you used DJI Mavic. For sure you didn't used this drone randomly. Please provide some observations, technical and application-oriented reasons why you picked this one.

Response 4: Thank you for bringing this to our attention. We used this drone due to its maximum flight-time and long-range capabilities, along with the possibility to set the downlink frequency and the possibility to record the GPS flight data in log files. These aspects are now described in lines 463-471 of the revised manuscript:

“This drone is a popular commercial drone and is a suitable choice for our experiments for several reasons. It has a maximum flight time of up to 45 minutes and a maximum transmission range of up to 15km (FCC) in LOS, which allow us to perform a high number of measurements in various points across a large area in a single flight. The drone uses the 2.4GHz and the 5.8GHz ISM frequency bands with user-configurable DL frequency and bandwidth, which allows us to set the target frequency and match it with the phase calibration signal. Moreover, the drone provides log files with GPS flight data, which allows us to compute the actual AoA of the drone and use it as a reference when evaluating the system performance.”

Comments 5: In the conclusions you mention that "Having a precise North reference of the array is important ". Could you also indicate some possible ways that could be achieved or researched in the future work ? 

Response 5: Thank you for pointing this out. In our approach, we oriented the UCA and the Narda ADFA towards the geographical North using the site map and physical references in the area. For future implementations, we intend to be more rigorous in our approach by using a compass and accounting for the magnetic declination, and/or using a professional GPS receiver to properly orient the UCA. The conclusion paragraph has been updated to include these ideas, in lines 555-563 of the revised manuscript:

“In our implementation, the 0° angle points towards the geographical North. As in many DF systems, this aligns with the concept of azimuth, where angles are measured clockwise from the North direction in a horizontal plane. To achieve this, a compass can be used to properly align the UCA, but magnetic declination should be accounted for to obtain accurate results. A GPS receiver can also be used to determine the position and the orientation of the UCA. Moreover, once the UCA is properly aligned, the antennas should be anchored for proper operation in windy conditions. For future work, this aspect will be more rigorously addressed to remove the positioning errors and offset caused by array misalignment.”

4. Response to Comments on the Quality of English Language

The reviewer comment on the quality of the English language is:

(x) English language fine. No issues detected  

5. Additional clarifications

In the revised manuscript, the changes highlighted in green are in response to the comments of reviewer 2. Any additional changes in the manuscript, such as renumbering of references, are highlighted in yellow.

Reviewer 2 Report

Comments and Suggestions for Authors

1. First of all, the authors have overlooked a vast of literature on UAV localization. A very poor literature review is done, which shows that the authors are not aware of the state-of-art techniques on UAV localization. Below are example references that the authors have overlooked, which are closely related to the authors' investigation (not fully listed out):

[1]. Assistant Vehicle Localization Based on Three Collaborative Base Stations via SBL-Based Robust DOA Estimation [J], IEEE IOTJ, 2019;

[2]. Multi-UAV Cooperative Localization for Marine Targets Based on Weighted Subspace Fitting in SAGIN Environment [J], IEEE IOTJ, 2022;

[3]. BSBL-Based Auxiliary Vehicle Position Analysis in Smart City Using Distributed MEC and UAV-Deployed IoT [J], IEEE IOTJ, 2023;

[4]. Fast localizing for anonymous UAVs oriented toward polarized massive MIMO systems [J], IEEE IOTJ, 2023;

[5]. 3D Positioning method for anonymous UAV based on bistatic polarized MIMO radar [J], IEEE IOTJ, 2023;

[6]. 2D-DOA estimation auxiliary localization of anonymous UAV using EMVS-MIMO radar [J], IEEE IOTJ, 2024;

2. The novelty of this work should be highlighted. Why UCA and why MUSIC are necessary? These reasons should be explained. 

3. The experment results lack theorical analyses. Those results should be compared with the theorical ones.

4. All the symbols should be explained in the context, e.g., the superscript T in (3), the superscript H in (6), et al.

5. Formart of references should be unifined.

Author Response

Response to Reviewer 2 Comments

1. Summary

We sincerely thank the reviewer for taking the time to analyze our work and provide us with valuable advice on how to improve our research. Please find the detailed responses below and the corresponding revisions/corrections highlighted in green in the revised manuscript.

2. Questions for General Evaluation

Reviewer’s Evaluation

Response and Revisions

Does the introduction provide sufficient background and include all relevant references?

Yes

Are all the cited references relevant to the research?

Can be improved

Please see the point-by-point response section below

Is the research design appropriate?

Can be improved

Please see the point-by-point response section below

Are the methods adequately described?

Yes

Are the results clearly presented?

Yes

Are the conclusions supported by the results?

Yes

3. Point-by-point response to Comments and Suggestions for Authors

Comments 1: First of all, the authors have overlooked a vast of literature on UAV localization. A very poor literature review is done, which shows that the authors are not aware of the state-of-art techniques on UAV localization. Below are example references that the authors have overlooked, which are closely related to the authors' investigation (not fully listed out):

[1]. Assistant Vehicle Localization Based on Three Collaborative Base Stations via SBL-Based Robust DOA Estimation [J], IEEE IOTJ, 2019;

[2]. Multi-UAV Cooperative Localization for Marine Targets Based on Weighted Subspace Fitting in SAGIN Environment [J], IEEE IOTJ, 2022;

[3]. BSBL-Based Auxiliary Vehicle Position Analysis in Smart City Using Distributed MEC and UAV-Deployed IoT [J], IEEE IOTJ, 2023;

[4]. Fast localizing for anonymous UAVs oriented toward polarized massive MIMO systems [J], IEEE IOTJ, 2023;

[5]. 3D Positioning method for anonymous UAV based on bistatic polarized MIMO radar [J], IEEE IOTJ, 2023;

[6]. 2D-DOA estimation auxiliary localization of anonymous UAV using EMVS-MIMO radar [J], IEEE IOTJ, 2024;

Response 1: Thank you for your advice and the insightful reference list. In the state-of-art we focused on practical implementations of RF solutions for UAV localization with measurement campaigns done in real environments, overlooking purely theoretical localization methods/optimizations, and in doing so we missed some recent valuable developments in this domain. Thanks to your suggestions, the revised manuscript is enriched with a more comprehensive state of art that contains the EVS MIMO radar solution (references [18-20]). Moreover, we have included a set of studies (references [21-24]) that consider more complex mathematical models to better simulate the environment/behavior of the system and propose AoA optimization methods, thus providing a bridge between classic numerical simulations and practical implementations in real environments. The new references are included in the state-of-art chapter in:

lines 176-182:

“Another state-of-art solution is the use of MIMO radars employing electromagnetic vectors sensors (EVS) [18-20] for 2D and 3D UAV localization. The work in [20] proposes the implementation of a monostatic EVS MIMO radar, while [19] models a bistatic configuration. In each case, an EVS is a collocated, complete polarized sensor system of three electric dipoles and three magnetic loops. The numerical simulation results show that a bistatic configuration is more reliable. Moreover, these approaches are insensitive to the geometry of the TX/RX array.”

And lines 199-217:

“Extensive work is done to develop more complex mathematical models to better simulate the environment/behavior of the system and to propose new methods to improve AoA performance such as signal sparce recovery (SSR), L1 – singular value decomposition (SVD), nuclear norm optimization and sparse Bayesian learning (SBL), and optimal weighted subspace fitting (WSF). [21] In [22] the authors model a complex practical environment including non-uniform noise and off-grid error and propose an assistant vehicle localization method based on SBL-based robust DoA estimation and three collaborative base stations with MIMO arrays. Extensive simulations show that the proposed method provides superior localization results. Another complex model is presented in [21]. Here, a multi-UAV cooperative localization system for marine targets is proposed. As each UAV is equipped with a monostatic MIMO radar, the model considers the unknown mutual coupling effect and provides a robust weighted block sparse reconstruction DOA estimation method based on optimal WSF. In [23], the authors consider the mutual coupling and the reduced computational power of IoT sensors and propose a framework composed of distributed mobile-edge computing and IoT to realize auxiliary vehicle position analysis, and develop a suitable block SBL algorithm for DF.

While simulations provide valuable insights and initial assessments, it's crucial to acknowledge the inherent gap between simulation and reality when assessing and optimizing DF solutions.”

Comments 2: The novelty of this work should be highlighted. Why UCA and why MUSIC are necessary? These reasons should be explained. 

Response 2: Thank you for bringing this to our attention. The state-of-art presents multiple practical implementations of UAV localization solutions, each having advantages and disadvantages. In our approach, we have chosen the MUSIC algorithm for AoA estimation because it is a super resolution algorithm and can achieve localization accuracy of up to sub degree. Moreover, to ensure non-ambiguous target localization in a full 360⁰ range a uniform circular array (UCA) geometry was required. The novelty of the implemented solution and approach is highlighted as contributions in the revised manuscript, in lines 223-237, as follows:

The novelty of our work consists in the particular practical implementation and the validation of the proposed solution. As such, the main contributions of this study are:

·       Implementation of the MUSIC algorithm on an SDR-based system using a five-element UCA. MUSIC is a super resolution algorithm and can achieve localization accuracy of up to sub degree. The use of a UCA with odd number of elements ensures a non-ambiguous target localization in a full 360⁰ range.

·       Evaluation of the proposed solution capability to perform full azimuth (360-degree) RF signal localization. This is done in a close-range setup, in a controlled environment using a dummy target signal.

·       Performance evaluation in a relevant outdoor environment, in a long-range setup, for ranges up to 2.5km using a drone as target.

·       Validation of the results by comparison with a professional DF solution (i.e., Narda ADFA) and by cross-referencing with the actual target AoA determined from the drone GPS coordinates.

Moreover, the benefits of choosing UCA over ULA are detailed in the revised manuscript, in lines 307-321, as follows:

“First, in a ULA setup, the orientation determines the reference direction, therefore it cannot distinguish between signals arriving from the front and those arriving from the back the array (front-to-back ambiguity). Secondly, the ULA configuration may exhibit inaccuracies in the end-fire region, when the target is in line with the elements of the array leading to unsufficient angular diversity (end-fire ambiguity). As a result, the algorithm may produce erroneous results or fail to provide reliable direction estimates. Thirdly, a ULA array can only evaluate the AoA in azimuth.

To overcome the limitations of ULA, a UCA configuration can be used. The circular geometry offers multiple advantages. First, the UCA can cover the entire azimuthal plane (full 360° range), ensuring that targets can be ocalized regardless of their angle of arrival relative to the array. Secondly, the UCA does not suffer from front-to-back and end-fire region ambiguity, providingunambiguous direction estimation. Thirdly, UCA can be used to estimate both azimuth and elevation at the same time, by careful consideration of the array design and signal processing.”

Comments 3:. The experment results lack theorical analyses. Those results should be compared with the theorical ones.

Response 3: Thank you for your valuable advice. When analysing the results, besides comparing the performance with the professional DF system, we also aimed to check three main theoretical aspects:
(1) the MUSIC pseudospectrum offers a general view on the number of signal sources and their respective positions. In practice,besides the peak corresponding to the target signal, the pseudospectrum also shows additional peaks, indicating interfering sources/reflections. This aspect is highlighted in the revise document in lines 417-426:

“Unlike the simulated pseudospectrum which exhibited high, narrow peaks only for the target sources, the MUSIC pseudospectrum obtained during the measurement campaigns is slightly different. Although there is only source that is accounted for (our target signal) the pseudospectrum shows several peaks of various amplitudes. A five-element array can indicate up to four target AoAs (number of array elements – 1). Here, the highest peak corresponds to the dummy target (AoA of 38°) while the other peaks indicate interfering sources or reflections. These differences occur because numerical simulations consider idealized conditions while real-world experiments are affected by the inherent complexities and uncertainties of the environment. Noise, variability of the propagation environment, and unknown interference sources can significantly impact the results.”

(2) the UCA geometry provides unambiguous results in a full 360° azimuth range. The experimental results confirmed these expectations, as indicated in the revised manuscript, lines 439-448:

As per theoretical expectations, the AoA results for the 36 positions show that the proposed system, using MUSIC and the five-element UCA configuration, can be used for 360° AoA localization of a target. Due to the UCA geometry, and odd number of array elements, the results show no front-to-back and end-fire region ambiguities. It should also be noted that the resolution of the scan angles is one degree, meaning that the resolution of the AoA determination is also one degree. As such, the system can clearly distinguish between the 36 target positions. The overall average error (with respect to the position of the target) is 3,2713°. In particular, the highest error is 7,8354° (target position at 10°). The lowest error is 0,1673° (target position at 290°). A possible source of error is the inaccuracy of the physical placement of the target.

(3) the AoA accuracy is influenced by the SNR. The short-range and long-range experiments proved this theoretical construct as well. This aspect is included in the revised manuscript in lines 498-504:

“Compared to the short-range scenario, the long-range scenario had more erroneous results, proving the theoretical construct that a low SNR decreases the accuracy of the AoA estimation. In the short-range scenario the SNR is almost constant, being controlled as the target signal is a continuous wave of constant amplitude and because the distance is constant (3m). However, in the long-range scenario the SNR varied uncontrollably because of the changing distance (125m – 2500m) and the drone’s adaptive power feature.”

Comments 4: All the symbols should be explained in the context, e.g., the superscript T in (3), the superscript H in (6), et al.

Response 4: Thank you for bringing this to our attention. These symbols are now explained in the revised manuscript:

line 256: “where symbol T stands for the transpose operation”

and lines 266-267: “where symbol H represents the Hermitian operator and E is the expectation operator”

Comments 5: Formart of references should be unifined.

Response 5: Thank you for pointing this out. We have checked the reference list and formatted the entries using the required MDPI/Sensors style. The update reference list is in the revised manuscript between lines 582-660:

1.       Rai, P. K.; Idsøe, H.; Yakkati, R. R.; Kumar, A.; Khan, M. Z. A.; Yalavarthy, P. K.; Cenkeramaddi, L. R. Localization and activity classification of unmanned aerial vehicle using mmWave FMCW radars. IEEE Sensors Journal, 2021, 21(14), 16043-16053. https://doi.org/10.1109/JSEN.2021.3075909

2.       Azari, M. M.; Sallouha, H.; Chiumento, A.; Rajendran, S.; Vinogradov, E.; Pollin, S. Key technologies and system trade-offs for detection and localization of amateur drones. IEEE Communications Magazine, 2018. 56(1), 51-57. https://doi.org/10.1109/MCOM.2017.1700442

3.       Kaleem, Z.; Rehmani, M. H. Amateur drone monitoring: State-of-the-art architectures, key enabling technologies, and future research directions. IEEE Wireless Communications, 2018, 25(2), 150-159. https://doi.org/10.1109/MWC.2018.1700152

4.       Abeywickrama, S.; Jayasinghe, L.; Fu, H., Nissanka, S.; Yuen, C. RF-based direction finding of UAVs using DNN. IEEE International Conference on Communication Systems (ICCS), Chengdu, China, December 2018. https://doi.org/10.1109/ICCS.2018.8689177

5.       Shi, X.; Yang, C.; Xie, W.; Liang, C.; Shi, Z.; Chen, J. Anti-drone system with multiple surveillance technologies: Architecture, implementation, and challenges. IEEE Communications Magazine, 2018, 56(4), 68-74. https://doi.org/10.1109/MCOM.2018.1700430

6.       Kang, H.; Joung, J.; Kim, J.; Kang, J.; Cho, Y. S. Protect your sky: A survey of counter unmanned aerial vehicle systems. IEEE Access, 2020, 8, 168671-168710. https://doi.org/10.1109/ACCESS.2020.3023473

7.       Güvenç, I.; Ozdemir, O.; Yapici, Y.; Mehrpouyan, H.; Matolak, D. Detection, localization, and tracking of unauthorized UAS and jammers. In 2017 IEEE/AIAA 36th Digital Avionics Systems Conference (DASC), St. Petersburg, FL, USA, September 2017. https://doi.org/10.1109/DASC.2017.8102043 

8.       Martelli, T.; Murgia, F.; Colone, F.; Bongioanni, C.; Lombardo, P. Detection and 3D localization of ultralight aircrafts and drones with a WiFi-based passive radar, In International Conference on Radar Systems (Radar 2017), Belfast, October 2017. https://doi.org/10.1049/cp.2017.0423

9.       Caris, M.; Johannes, W.; Sieger, S.; Port, V.; Stanko, S. Detection of small UAS with W-band radar. In 2017 18th International Radar Symposium (IRS), Prague, Czech Republic, June 2017. https://doi.org/10.23919/IRS.2017.8008143

10.    Stojkoska, B. R.; Palikrushev, J.; Trivodaliev, K.; Kalajdziski, S. Indoor localization of unmanned aerial vehicles based on RSSI. In IEEE EUROCON 2017-17th International Conference on Smart Technologies, Ohrid, Macedonia, July 2017. https://doi.org/10.1109/EUROCON.2017.8011089

11.    Morris, P. J. B.; Hari, K. V. S. Detection and localization of unmanned aircraft systems using millimeter-wave automotive radar sensors. IEEE Sensors Letters, 2021, 5(6), 1-4. https://doi.org/10.1109/LSENS.2021.3085087

12.    Bisio, I.; Garibotto, C.; Haleem, H.; Lavagetto, F.; Sciarrone, A. On the localization of wireless targets: A drone surveillance perspective. IEEE Network, 2021, 35(5), 249-255. https://doi.org/10.1109/MNET.011.2000648

13.    Nguyen, P.; Kim, T.; Miao, J.; Hesselius, D.; Kenneally, E.; Massey, D., Vu, T.. Towards RF-based localization of a drone and its controller. In Proceedings of the 5th workshop on micro aerial vehicle networks, systems, and applications, Seoul Republic of Korea, June 2019. https://doi.org/10.1145/3325421.3329766

14.    Meles, M.; Mela, L.; Rajasekaran, A.; Ruttik, K.; Jäntti, R. Drone localization based on 3D-AoA signal measurements. In 2022 IEEE 95th Vehicular Technology Conference (VTC2022-Spring), Helsinki, Finland, June 2022. https://doi.org/10.1109/VTC2022-Spring54318.2022.9860965

15.    Rice, T.; Pandey, D.; Ramirez, D.; Knightly, E. Experimental Evaluation of AoA Estimation for UAV to Massive MIMO. In MILCOM 2023-2023 IEEE Military Communications Conference (MILCOM), Boston, MA, USA, October 2023. https://doi.org/10.1109/MILCOM58377.2023.10356267

16.    Kaplan, B.; Kahraman, İ.; Ektı, A. R.; Yarkan, S.; Görçın, A.; Özdemır, M. K.; Çirpan, H. A. Detection, identification, and direction of arrival estimation of drone FHSS signals with uniform linear antenna array. IEEE Access, 2021, 9, 152057-152069. https://doi.org/10.1109/ACCESS.2021.3127199

17.    Sacco, G.; Pittella, E.; Pisa, S.; Piuzzi, E. A MISO radar system for drone localization. In 2018 5th IEEE International Workshop on Metrology for AeroSpace (MetroAeroSpace), Rome, Italy, June 2018). https://doi.org/10.1109/MetroAeroSpace.2018.8453572

18.    Wen, F. et al. Fast Localizing for Anonymous UAVs Oriented Toward Polarized Massive MIMO Systems. IEEE Internet of Things J, 2023, 10 (22), 20094-20106. https://doi.org/10.1109/JIOT.2023.3282644

19.    Wen, F.; Shi, J.; Gui, G.; Gacanin, H.; Dobre, A. 3-D Positioning Method for Anonymous UAV Based on Bistatic Polarized MIMO Radar. IEEE Internet of Things J., 2023, 10(1), 815-827. https://doi.org/10.1109/JIOT.2022.3204267

20.    Wen, F.; Zhang, Z.; Sun, H.; Gui, G.; Sari, H.; Adachi, F. 2D-DOA Estimation Auxiliary Localization of Anonymous UAV Using EMVS-MIMO Radar. IEEE Internet of Things J., 2024. https://doi.org/10.1109/JIOT.2024.3351136

21.    Wang, X.; Yang, L. T.; Meng, D.; Dong, M.; Ota, K.; Wang, H. Multi-UAV Cooperative Localization for Marine Targets Based on Weighted Subspace Fitting in SAGIN Environment. IEEE Internet of Things J., 2022, 9(8), 5708-5718. https://doi.org/10.1109/JIOT.2021.3066504

22.    Wang, H.; Wan, L.; Dong, M.; Ota, K.; Wang, X. Assistant Vehicle Localization Based on Three Collaborative Base Stations via SBL-Based Robust DOA Estimation. IEEE Internet of Things J., 2019, 6(3), 5766-5777. https://doi.org/10.1109/JIOT.2019.2905788

23.    Wang, H.; Wang, X.; Lan, X.; Su, T.; Wan, L. BSBL-Based Auxiliary Vehicle Position Analysis in Smart City Using Distributed MEC and UAV-Deployed IoT. IEEE Internet of Things J., 2023, 10(2), 975-986. https://doi.org/10.1109/JIOT.2022.3204986

24.    Schmidt, R. Multiple emitter location and signal parameter estimation. IEEE Transactions on Antennas and Propagation,1986, 34(3), 276-280. https://doi.org/10.1109/TAP.1986.1143830

25.    Gupta, P.; Kar, S. P. MUSIC and improved MUSIC algorithm to estimate direction of arrival. In 2015 International Conference on Communications and Signal Processing (ICCSP), Melmaruvathur, India, 2015. https://doi.org/10.1109/ICCSP.2015.7322593

26.    MUSIC Super-Resolution DOA Estimation. Available online: https://www.mathworks.com/help/phased/ug/music-super-resolution-doa-estimation.html#bvd8ug3, (accessed on 5 February 2024).

27.    Xie, J.; He, Z.; Li, H.; Li, J. 2D DOA estimation with sparse uniform circular arrays in the presence of mutual coupling. EURASIP Journal on Advances in Signal Processing, 2011, 127, 1-18. https://doi.org/10.1186/1687-6180-2011-127

28.    USRP-2954 Specifications. Available online: https://www.ni.com/docs/en-US/bundle/usrp-2954-specs/page/specs.html, (accessed on 5 February 2024).

29.    CDA-2990 Specification, Available online: https://www.ni.com/docs/en-US/bundle/cda-2990-specs/page/specs.html, (accessed on 5 February 2024).

30.    HM8135 3GHz RF-Synthesizer User Manual. Available online: https://cdn.rohde-schwarz.com/pws/dl_downloads/dl_common_library/dl_manuals/gb_1/h/hm8135_x/HM8135_UserManual_de_en_04_1.pdf, (accessed on 5 February 2024).

31.    Van der Merwe, J. R.; Malan, J.; Maasdorp, F. D. V.; Du Plessis, W. P. Multi-channel software defined radio experimental evaluation and analysis. Proc. Instituto Tecnologico de Aeronautica (ITA), September 2014. http://hdl.handle.net/10204/7778

32.    2.4/5.8 GHz 3.5/4 dBi Dual Band Omnidirectional Antenna - N-Female Connector. Available online: https://www.l-com.com/wireless-antenna-24-58-ghz-35-4-dbi-dual-band-omnidirectional-antenna-n-female-connector, (accessed on 10 March 2024).

33.    DJI Mavic 3T-Enterprise, Available online: https://enterprise.dji.com/mavic-3-enterprise/specs (accessed on 11 April 2024)

34.    Narda SignalShark Handheld ADFA, Automatic DF Antenna. Available online: https://www.narda-sts.com/en/signalshark-handheld/adfa-2-df-antenna/ (accessed on 10 March 2024).

4. Response to Comments on the Quality of English Language

The reviewer comment on the quality of the English language is:

(x) I am not qualified to assess the quality of English in this paper

5. Additional clarifications

In the revised manuscript, changes highlighted in blue are in response to the comments of reviewer 1. Any additional changes in the manuscript, such as renumbering of references, are highlighted in yellow.

Round 2

Reviewer 2 Report

Comments and Suggestions for Authors

All my concerns have been stressed.